# N-ForGOT: Towards Not-forgetting and Generalization of Open Temporal Graph Learning

**Liping Wang**[1] , **Xujia Li**[2*], **Jingshu Peng**[2], **Yue Wang**[2], **Chen Zhang**[3], **Yan Zhou**[3], **Lei Chen**[1,2]
[1]The Hong Kong University of Science and Technology (Guangzhou)
[2]The Hong Kong University of Science and Technology  [3] Chuanglin Tech

## Abstract

Temporal Graph Neural Networks (TGNNs) lay emphasis on capturing node interactions over time but often overlook evolution in node classes and dynamic data distributions triggered by the continuous emergence of new class labels, known as the open-set problem. This problem poses challenges for existing TGNNs in preserving learned classes while rapidly adapting to new, unseen classes. To address this, this paper identifies two primary factors affecting model performance on the open temporal graph, backed by a theoretical guarantee: (1) *the forgetting of prior knowledge* and (2) *distribution discrepancies between successive tasks*. Building on theoretical insights, we propose **N-ForGOT**, which incorporates two plug-in modules into TGNNs to preserve prior knowledge and enhance model generalizability for new classes simultaneously. The first module preserves previously established inter-class connectivity and decision boundaries during the training of new classes to mitigate the forgetting caused by temporal evolutions of class characteristics. The second module introduces an efficient method for measuring distribution discrepancies with designed temporal Weisfeiler-Lehman subtree patterns, effectively addressing both structural and temporal shifts while reducing time complexity. Experimental results on four public datasets demonstrate that our method significantly outperforms state-of-the-art approaches in prediction accuracy, prevention of forgetting, and generalizability[1].

## 1 Introduction

Temporal graph neural networks (TGNNs) are increasingly recognized for their ability to encapsulate temporal interactions among nodes in temporal graphs (Shi et al., 2018; Kumar et al., 2019b; Wang et al., 2023). Despite their substantial contributions, many current TGNNs (Rossi et al., 2020; Cong et al., 2023) assume static data distribution and unchanging class sets, neglecting the reality that new categories frequently emerge in real-world scenarios. For instance, new topics continually emerge into the topic community graph (Feng et al., 2023; Hamilton et al., 2017), where nodes represent post and are labeled by their topics. This necessitates the development of TGNN models that accommodate open temporal graphs (OTGs), adapting to continuously expanding class sets and evolving interactions.

As illustrated in Fig. 1, Open Temporal Graph Learning (OTGL) is conceptualized as a series of chronological tasks, each introducing previously unseen classes that can lead to shifts of existing data distribution (Feng et al., 2023). However, existing methods treat emerging unseen data as potential disruption, focusing solely on preserving previously learned knowledge, which undermines the model's performance in generalizing both historical and incoming data.

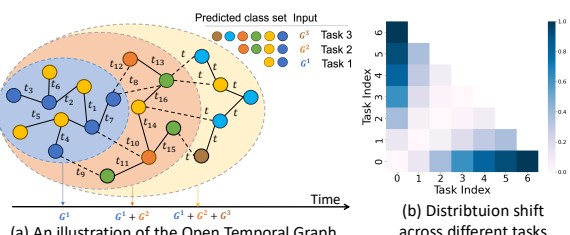

(a) An illustration of the Open Temporal Graph

(b) Distribtuion shift across different tasks

Figure 1: Open Temporal Graph Learning (OTGL): class evolution and distribution shifts

---

*Corresponding Authors: Xujia Li, leexujia@ust.hk

[1]Code is available at `https://github.com/Liane-Wang/N-ForGOT`

Motivated by the need to balance knowledge preservation from existing data with the swift adaptation to incoming unseen classes. We theoretically analyzed the generalization error of the OTGL and investigated two primary factors: *1) forgetting of prior knowledge and 2) distribution discrepancies between successive tasks.*

**Forgetting of prior knowledge.** Catastrophic forgetting poses a significant challenge in OTGs, which continuously evolve with new elements and structures (Feng et al., 2023). In OTGs, the irregularity of node neighborhoods and the ongoing introduction of new edges disrupt existing node correlations, complicating continual learning (Lu et al., 2022; Yuan et al., 2023). Recent research has tried to address catastrophic forgetting within graph-structured data, yet most studies focus on static graph snapshots (Wang et al., 2022a; Zhang et al., 2022a), missing temporal topological information crucial for effectively managing forgetting in OTGs. Some works (Feng et al., 2023) aimed to preserve temporal topological information by sampling triad motifs from temporal graphs and replaying them for continual learning. However, the invariant structure of triad motifs struggles to represent class characteristics that dynamically change over time. For instance, the evolution of cell phones in e-commerce networks from single to dual-screen and foldable models illustrates the *temporal inconsistencies*. This evolution highlights that class characteristics change over time, exacerbating the preservation of previously learned information and intensifying catastrophic forgetting in OTGs.

**Distribution discrepancies between tasks.** To achieve broad generalization across both historical and emerging data distributions, it is essential for OTGL to adeptly navigate the parameter space by measuring the discrepancy between task distributions. Quantifying task discrepancies in OTGs can arise from multiple domains, such as *structure shifts*, which arise from the differences in graph structures between old and new data and *time shifts*, caused by the continuous evolution of the OTG over time (Zhang et al., 2022b). Fig. 1(b) highlights high distribution shifts (blue grid) between the graph introduced by different tasks. Prior studies have utilized integral Maximum Mean Discrepancy-based approaches (MMD) (Gretton et al., 2012; Zellinger et al., 2017) to measure structure shifts at the graph representation-level (Zhu et al., 2021; Zheng et al., 2024). However, the computational demands of these approaches, which scale at least $\mathcal{O}(n^2)$ (Bernton et al., 2019; Gretton et al., 2012) relative to the graph size, pose significant challenges for TGNNs. These networks, known for their *limited efficiency*, heavily rely on batch training to enhance processing (Rossi et al., 2020; Li et al., 2023). Although batch processing facilitates parallel processing within batches, it often neglects the temporal dependencies across different batches (Su et al., 2024), leading to inaccuracies in measuring distribution shifts compared to methods that assess the entire graph. This oversight leads to two issues: measuring discrepancies between entire graphs becomes infeasible, decelerating the training process; batch-wise training can reduce the accuracy of the discrepancy measurements due to the isolated nature of each batch.

Motivated by these challenges in OTGL, we explore the OTGL from a new perspective by going beyond merely addressing forgetting. We propose a novel OTGL approach, Towards **N**ot-**For**getting and **G**eneralization of **O**pen **T**emporal Graph (**N-ForGOT**), including two plug-in modules for TGNNs: a temporal interclass connectivity regularization module (TICR) to minimize forgetting of prior knowledge and a localized temporal graph discrepancy optimization module (LTDO) to address task discrepancies. To mitigate catastrophic forgetting, the TICR module renormalizes parameters to align new data with previous tasks, *preserving inter-class connectivity and decision boundaries*. Through this, TICR maintains essential class features to prevent overfitting. To enhance the generalizability, the LTDO module offers an *efficient strategy for measuring multi-domain distribution shifts* using a designed local structure alongside a linear approximation of MMD. We develop a temporal Weisfeiler-Lehman (WL) subtree pattern, which is capable of capturing structural and temporal information across different batches. This capability can effectively measure distribution shifts during batch training while reducing the information loss associated with batch data isolation. LTDO module significantly reduces the computational complexity from $\mathcal{O}(n^2)$ to $\mathcal{O}(n)$ and lowers the storage complexity from $\mathcal{O}(n)$ to $\mathcal{O}(1)$ (as elucidated in Sec. 4.2).

Our main contributions can be summarized as: 1) We extend beyond the conventional focus on mitigating forgetting, advancing towards a balance between forgetting and generalizability in the OTGL. To achieve this, we propose two plug-in modules for TGNNs; 2) We introduce the TICR module to mitigate the forgetting of prior knowledge by leveraging the interconnectivity of the graph; 3) We introduce the LTDO module with a novel temporal WL subtree patterns representation, reducing computational complexities while enhancing model generalizability.

## 2 RELATED WORK

**TGNNs.** Temporal graphs represent sequences of time-stamped events. According to the sampling scheme for aggregation process, TGNNs can be categorized into two categories: neighborhood-based message-passing TGNNs (MP-TGNNs) (Xu et al., 2020; Rossi et al., 2020; Kumar et al., 2019a; Trivedi et al., 2019), which aggregate the node's temporal neighborhoods information, and walk-aggregating TGNNs (Souza et al., 2022; Wang et al., 2021), which encode temporal walks from the target node. Additionally, memory-updated TGNNs (Rossi et al., 2020; Li & Chen, 2023) build on the MP-TGNN framework to incorporate a memory state vector that stores historical node features, updating embeddings as relevant events occur. However, these models overlook information loss during updates and struggle with new, unseen node classes.

**Graph Continual Learning.** OTGL stems from Graph Continual Learning (GCL) but is specifically designed to tackle the unique challenges of temporal graphs. Recent studies in graph continual learning have aimed to address catastrophic forgetting in graph-structured data, which can be categorized into two types: regularization-based and replay-based methods. Regulation-based methods focus on maintaining old data distributions by selectively constraining updates to parameters that are crucial for previous task performances. Liu et al. (2021) evaluated both parameter gradients and topological feature gradients to identify and preserve parameters vital for maintaining key topological information. Replay-based methods (Zhou & Cao, 2021; Wang et al., 2022a; Feng et al., 2023), stored portions of historical data in a memory buffer to help preserve previously learned knowledge. They utilized various sampling strategies or data generation techniques (Wang et al., 2022a) to replay historical graph structures, including walk sequences (Wang et al., 2022a), node sets (Zhou & Cao, 2021), neighborhood sets, and different motif structures (Feng et al., 2023), to maintain learned knowledge. DeLoMe (Niu et al., 2024) and CaT (Liu et al., 2023) employ condensed graphs as replay data. These condensed graphs are significantly smaller than the original, yet they consist of synthetic node representations that preserve the complete informational content of the original structures. Innovatively, OTGNet (Feng et al., 2023), the only work addressing forgetting in OTGs, employed triad sampling to preserve key structural evolution information from historical data. However, the effectiveness of OTGNet is heavily dependent on the volume of replayed data. Insufficient data buffers can lead to suboptimal results, while increased sampling could significantly prolong the overall training time. In contrast to these methods, which view updates from new tasks as potential disruptions to previously learned knowledge, our approach seeks a balance between retaining knowledge from past data and effectively adapting to new, evolving data distributions.

## 3 PRELIMINARY

In this section, we illustrate preliminary definitions of OTG and the discrepancy metric and provide a theoretical analysis of the generalization error bound within the OTGL.

**Problem Definition.** In the open temporal graph setting (Feng et al., 2023), the graph dynamically evolves with the continuous addition of new nodes, leading to the emergence of new classes and, correspondingly, new node classification tasks. Consider a temporal graph $\mathbf{G} = (V, T, \mathcal{E})$, where $V$ is a set of nodes, $T \subseteq \mathbb{N}_0$ is a finite set of timestamps, and $\mathcal{E}$ comprises temporal edges. These edges are defined as the set $(u, v, t)$ for all node pairs $u, v \in V$ and timestamps $t \in T$ where an edge exists between nodes $u$ and $v$ at time $t$. The open temporal graph is conceptualized as a sequence of tasks $\mathcal{K} = \{\mathcal{K}_1, ..., \mathcal{K}_k\}$, arranged in chronological order, where $k$ represent the current task. The task can be represented as $\mathcal{K}_i = \{(\mathbf{G}_i, \mathbf{Y}_i)\}$, containing a temporal graph $\mathbf{G}_i$ and a corresponding class set $\mathbf{Y}_i$. Each task $\mathcal{K}_i$ introduces new classes that were not present in earlier tasks, and $\mathbf{G}_i$ contains data specific to the timeslice of task $\mathcal{K}_i$.

**Performance Bound Analysis of OTGL.** The objective of OTGL is to develop a TGNN, $f$, parameterized by $\theta$, that incrementally learns across a sequence of $k$ tasks. We decompose the trained TGNN $f = g \circ h$ [2] into the feature extractor $g : \mathbf{G} \to \mathbb{R}^{d'}$ ($Z = g(\mathbf{G})$) and discriminator $h : \mathbb{R}^{d'} \to \mathbf{Y}$ ($\mathbf{Y} = h(Z)$). Node embeddings are produced by $Z(i) = g(x_i | x_i \in V)$. These embeddings are subsequently processed by the discriminator. To formulate the expected loss of the model $f$, over the entire distribution of data, termed the *population risk*, we first defined the distribution for the current task as $\mathbb{G}_k$ and the aggregate historical data distribution as $\mathbb{G}_{1:k-1} := \{\mathbb{G}_i\}_{i=1}^{k-1}$. Formally, with a given loss function, $\ell(.,.) \to \mathbb{R}$, the population risk of the current task (Wang et al., 2022b) can be defined as $\mathcal{R}_{\mathbb{G}_k} = \mathbb{E}_{(G|V,Y) \sim \mathbb{G}_k}[\ell(f_k(\theta, G), y)]$. Likewise, the population risk over the distribution of previous tasks can be defined as $\mathcal{R}_{\mathbb{G}_{1:k-1}} = \frac{1}{k-1} \sum_{i=1}^{k-1} \mathbb{E}_{(G|V,Y) \sim \mathbb{G}_i}[\ell(f_i(\theta, G), y)]$.

---

[2] For simplicity, we use $\theta$ to represent the parameters of functions $g$, $h$ and $f$

To achieve a balance between forgetting and generalizability, our method aims to minimize the population risk for the current task, $\mathcal{R}_{\mathbb{G}_k}$, while also reducing the generalization error, defined as $|\mathcal{R}_{\mathbb{G}_k} - \mathcal{R}_{\mathbb{G}_{1:k-1}}|$. We analyze this generalization error based on PAC-Bayesian theory Li & Bilmes (2007) and establish a bound for $\mathcal{R}_{\mathbb{G}_k}$.

**Theorem 1** *(You et al., 2023; Li et al., 2021; Wang et al., 2022b) Let $\mathcal{H}$ be a hypothesis space of Vapnik-Chervonenkis dimension $d_c$, with probability at least $1 - \delta$ for any $\delta \in (0, 1)$. For the hypothesis function $f$, the population risk of the current task is bounded as follows (please see a complete proof in Appendix A):*

$$\mathcal{R}_{\mathbb{G}_k}(f_k(\theta_k)) \leq \hat{\mathcal{R}}_{\mathbb{G}_{1:k-1}}(f_k(\theta_k)) + div_{\mathcal{H}}\left(\mathbb{G}_{1:k-1}, \mathbb{G}_k\right) + \sqrt{\frac{2d}{m}\log(\frac{em}{d})} + \sqrt{\frac{1}{2m}\log\left(\frac{1}{\delta}\right)} + \xi,$$

*where $\xi = \min\{\mathrm{E}_{\mathbb{G}_k}\left[|f_{1:k-1}(\mathbf{x}) - f_k(\mathbf{x})|\right], \mathrm{E}_{\mathbb{G}_{1:k-1}}[|f_{1:k-1}(\mathbf{x}) - f_k(\mathbf{x})]\}$ is the difference in labeling functions across the new and old tasks, which we expect to be small. $div(\cdot)$ is a distance metric that formulates the old and new task distributions in a Reproducing Kernel Hilbert Space (RKHS).*

Motivated by Theorem 1, which is extended from the domain adaptation bound analysis (Li et al., 2021), we identify two components that bound the performance of OTGL: 1) The term $\hat{\mathcal{R}}_{\mathbb{G}_{1:k-1}}(f_k(\theta_k))$ indicates the model's performance on historical data distribution $\mathbb{G}_{1:k-1}$, which is determined by the model's ability in managing forgetting; 2) $div_{\mathcal{H}}(\mathbb{G}_{1:k-1}, \mathbb{G}_k)$ represents the discrepancy between task distributions, which can be quantified using the MMD. Accordingly, this work target on minimizing $\hat{\mathcal{R}}_{\mathbb{G}_{1:k-1}}(f_k(\theta_k))$ and the divergence $div_{\mathcal{H}}(\mathbb{G}_{1:k-1}, \mathbb{G}_k)$, leading to the corresponding proposal of two specific plug-in modules: TICR and LTDO.

Specifically, the first term, $\hat{\mathcal{R}}_{\mathbb{G}_{1:k-1}}(f_k(\theta_k))$, is an empirical measure that quantifies the risk associated with the model $f$ over the training data from previous tasks. This measure reflects how well the model performs on data from previous tasks by calculating the average loss across these historical data distributions. It indicates the model's performance on historical data $\mathbf{G}_{1:k-1}$, given that it was trained using the current data $\mathbf{G}_k$. Therefore, the value of the first term, $\hat{\mathcal{R}}_{\mathbb{G}_{1:k-1}}(f_k(\theta_k))$, reflecting the model's capacity to mitigate catastofic forgetting. Successfully evaluating this component hinges on a deep understanding of how historical data influences and constrains the parameter space. The second term $div_{\mathcal{H}}(\mathbb{G}_{1:k-1}, \mathbb{G}_k)$ quantifies the discrepancy between the distributions $\mathbb{G}_{1:k-1}$ and $\mathbb{G}_k$, which can be formulated by the MMD metric (detailed definition can found in Appendix C). MMD can be expressed as $\mathrm{MMD}^2(\mathbb{G}_{1:k-1}, \mathbb{G}_k) = \sup_{\|\phi\|_{\mathcal{H}} \leq 1} \left\|\mathbb{E}_{x \sim \mathbb{G}_{1:k-1}}[\psi(x)] - \mathbb{E}_{y \sim \mathbb{G}_k}[\psi(y)]\right\|_{\mathcal{H}}^2$, where $\psi$ is a kernel function that maps data from the two distributions to a RKHS.

## 4 Methlodogy

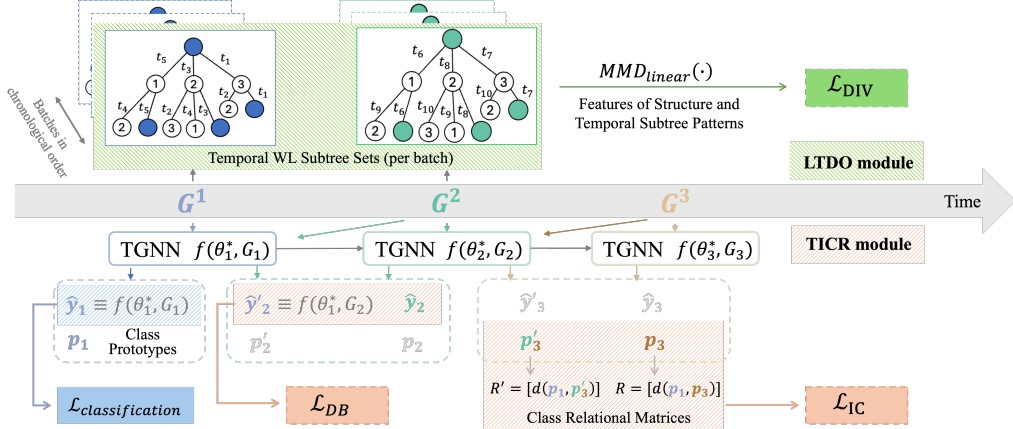

Figure 2: Overview of **N-ForGOT**.

The overall architecture of **N-ForGOT** is illustrated in Figure 2, including two plug-in modules: the Temporal Inter-class Connectivity Regularization (TICR) module (Sec. 4.1), highlighted in red, is designed to mitigate catastrophic forgetting; and the Localized Temporal Graph Discrepancy Optimization (LTDO) module (Sec. 4.2), shown in green, focuses on optimizing distribution discrepancies across successive tasks.

## 4.1 Temporal Interclass Connectivity Regularization (TICR) module

As the set of classes within the graph expands, performance on previous classes often deteriorates due to the forgetting of established class prototype knowledge. As introduced before, the temporal inconsistencies of class characteristics take more challenge to address the forgetting. To counteract this, we preserve knowledge through two key components: *decision boundaries* and *feature-based inter-class connectivity* among different classes. Decision boundaries and inter-class connectivity typically represent consistent knowledge. For example, despite the characteristics of the class 'cell phone' evolving with technological development, the difference between it and 'earphones' remains obvious and consistent over time. The model learns task-specific parameters according to these two components and renormalizes current ones to align with those from previous tasks.

**Decision Boundaries.** To preserve task-specific decision boundaries, we ensure that the network retains the essential properties of each class and contrasts features from the past with the current. Inspired by the modified cross-entropy loss function (Hinton et al., 2015), we apply logistic outputs from the old network serving as soft targets in the loss function for approximating the predicted class probabilities of the current network.

$$\mathcal{L}_{DB}(\hat{\mathbf{y}}_k, \hat{\mathbf{y}}'_k) = -\sum_{i=1}^{|Y_k|} \frac{(\hat{\mathbf{y}}'^{(i)}_k)^{1/\tau}}{\sum_j (\hat{\mathbf{y}}'^{(j)}_k)^{1/\tau}} log \left( \frac{(\hat{\mathbf{y}}^{(i)}_k)^{1/\tau}}{\sum_j (\hat{\mathbf{y}}^{(j)}_k)^{1/\tau}} \right) \tag{1}$$

where $\hat{\mathbf{y}}'_k \equiv f_{k-1}(\theta_{k-1}, \mathbf{G}_k)$ and $\hat{\mathbf{y}}_k \equiv f_k(\theta_k, \mathbf{G}_k)$ represent the logits from both previous and current parameter configurations of the class set $Y_c$ in the ongoing task. The loss function incorporates a temporal scaling function to adjust the logits, enhancing the model's discriminatory capabilities across temporal transitions, where $\tau$ is the temperature factor. The temperature parameter is set as $\tau > 1$ to amplify the influence of smaller logit values. This adjustment accounts for potential label imbalance by giving more weight to less frequent categories.

**Inter-class Connectivity.** To capitalize on interclass connectivity, we assess the distances between class relational matrices from networks trained on previous and current tasks. Prototypes for the class $i$ are denoted by $\mathbf{p}^{(i)} = \sum_{j \in \mathcal{C}_i} g(x_j)$, where $\mathcal{C}_i$ denotes the node set related to class $i$. After fully training the current task, prototypes for the newly involved classes are updated in the class prototype matrix, $\mathbf{P}_k = \left[ \mathbf{p}_k^{(1)}, \mathbf{p}_k^{(2)}, \ldots, \mathbf{p}_k^{(N_c)} \right]$, $\mathbf{p}_k^{(i)} = \sum_{v=1}^{k-1} I(i, v) \cdot \mathbf{p}_v^{(i)}$, $\forall n \in \mathbf{Y}_{1:k}$, where $N_c = |\mathbf{Y}_{1:k}|$ denotes the number of classes in the class set, encompassing all previous tasks, and $I(i, v)$ is an indicator function used to represent whether a category $i$ appears in a task $v$. This formulation establishes a dynamic and evolving class prototype matrix incorporating historical data to preserve and utilize the interclass relationships effectively as the class set expands.

Interclass connectivity of previous and current tasks can be formulated by class relational matrices, which are defined as:

$$R = \left[ d(\mathbf{p}_{k-1}^{(j)}, \hat{\mathbf{p}}^{(i)}) \right]_{i,j=1}^c, \ R' = \left[ d(\mathbf{p}_{k-1}^{(j)}, \hat{\mathbf{p}}'^{(i)}) \right]_{i,j=1}^c \tag{2}$$

$R$ and $R'$ are the class relational matrices of the current and previous networks respectively, where $d(\cdot, \cdot)$ denotes the Euclidean distance to measure the distance between class prototypes. Each matrix with dimensions $c \times c$, where $c = |\mathbf{Y}_k|$ represents the number of classes in the current task. $\mathbf{p}_{k-1}^{(j)}$ is obtained from the class prototype matrix which represents the ground truth features of previous classes retained after fully training each previous task. $\hat{\mathbf{p}}$ and $\hat{\mathbf{p}}'$ represent the class prototypes of new classes obtained from the current task and previous task networks, respectively. The class prototype $\hat{\mathbf{p}}'$ of class $i$ derived from the previous network is defined as:

$$\hat{\mathbf{p}}'^{(i)} = \frac{1}{|\mathcal{C}_i|} \Big| \sum_{j \in \mathcal{C}_i \sim \mathbf{G}_k} g_{k-1}(\theta_{k-1}, x_j) \tag{3}$$

where $\mathcal{C}_i$ represents the set of nodes in class $i$ within the graph of the current task. Similarly, the prototype set $\hat{\mathbf{p}}$ for the current network is computed using the current model parameters.

We define the interconnectivity class loss to align the class relational matrices between previous and current tasks:

$$\mathcal{L}_{IC} = \|R - R'\|_2 \tag{4}$$

This loss function minimizes the distance between class prototypes across models, ensuring consistent class understanding and reducing catastrophic forgetting of previous classes.

### 4.2 LOCALIZED TEMPORAL GRAPH DISCREPANCY OPTIMIZATION (LTDO) MODULE

This module addresses multi-domain distribution shifts and efficiency challenges in measuring discrepancies between task distributions. The LTDO module quantifies both structural and temporal shifts within open temporal graphs, identifying task variations based on the designed local structure. Leveraging the topology of this local structure, it enables efficient batch-wise task discrepancy measurement, while also minimizing information loss typically associated with isolated batches.

**Temporal WL Subtree Patterns.** Inspired by the capability of the WL subtree in addressing graph isomorphism challenges (Leman & Weisfeiler, 1968). We introduce the temporal WL subtree with structure and temporal patterns as the discriminative local structure, which makes it particularly suited for capturing nuanced structural differences in dynamic graph environments.

**Definition 1** *Temporal WL Subtree Patterns. For a temporal graph, $\mathbf{G} = (V, T, \mathcal{E})$. The neighborhood of a node $v \in V$ is denoted as $\mathcal{N}(v) = \{u \in V \mid (v, u, t) \in \mathcal{E}\}$. For a node $v$, its temporal subtree pattern $S^d(v) = (S_s^d, S_t^d)$ with depth $d$ contains structure and temporal parts. The temporal WL subtree can be defined recursively in the following way:*

$$S_s(v): \ S_s^{(d=0)}(v) = \{v\}, \ S_s^{(d>0)}(v) = S_s^{d-1}(v) \cup \{(u', \{\mathcal{N}(u')\}) \mid u' \in \hat{S}_s^{d-1}\};$$

$$S_t(v): \ S_t^{(d=0)}(v) = \{t'\}, \ S_t^{(d>0)}(v) = S_t^{d-1}(v) \cup \{|t' - t| \mid (u', u, t) \in \mathcal{E}, u' \in \hat{S}_s^{d-1}, u \in \hat{S}_s^d\}$$

*where $\hat{S}_v^d$ denote a set of the leaf nodes in $S_v^d$, and $t_{max}$ is the maximum observed timestamp of the target node $v$.*

We further develop the representation of temporal WL subtree patterns for MMD measurement by aligning the TGNN aggregation process with the WL algorithm (Leman & Weisfeiler, 1968). As delineated in Definition 1, the initial pattern $S^0(v)$ represents the categorical feature for the target node. Subsequent patterns, from $S^1(v)$ to the higher order, encapsulate the topological structure surrounding the target node, along with evolving temporal information. The subtree structure is defined recursively via node neighborhood, allowing us to utilize neighborhood-based message-passing TGNNs to capture the features of temporal subtree patterns. We proposed the following definition to formulate the structural and temporal features of our temporal WL subtree patterns.

**Definition 2** *With the feature extractor $g : \mathbf{G} \to \mathbb{R}^{d'}$ in the TGNN, which contains a time encoding function $\Phi : T \to \mathbb{R}^{d'_t}$. The depth of our temporal WL subtree refers to the number of TGNN layers. Considering a sufficient number of TGNN layers, the structure and temporal features for the temporal WL subtree patterns can be defined as (theoretical proof is provided in Appendix B):*

*a) Structure feature for $d^{th}$ pattern:*

$$C_s^d(v) = \tilde{g}^{(d)}(v) = AGG^{(d)} \left( \left\{ \left( g^{(d-1)}(u), e \right) \mid (u, e, t) \in S_s^d(v) \right\} \right), \quad (5)$$

*where AGG is layer-specific aggregation function. If $d = 0$, $C\left(S_s^0(v)\right)$ employs the raw feature of the target node $x_v$.*

*b) Temporal feature for $d^{th}$ pattern:*

$$C_t^d(v) = \Phi(\Delta t), \ where \ \Delta t = S_t^0(v) - t_i, \ t_i \in S_t^d(v)) \quad (6)$$

Definition 2 is based on the theorem that message-passing TGNNs with $d$ layers yield equivalent outputs for nodes that share the same temporal WL subtree pattern of depth $d$, when the underlying structures are non-isomorphic. This validates the compatibility of computing features for the temporal WL subtree patterns across various TGNN architectures. It also demonstrates that our LTDO module can effectively function as a plug-in module within TGNNs. For instance, the time encoder function in TGNNs enables the computation of temporal features in our temporal WL subtree. TGAT (Xu et al., 2020) employs a self-attention mechanism for message passing and incorporates a trainable time-encoding function $\Phi(\Delta t) = cos(t\mathbf{w} + \mathbf{b})$. Similarly, the GraphMixer (Cong et al., 2023), another variant of TGNN, utilizes a multi-layer perceptrons-based aggregation function combined with a simpler, untrained time encoder, $\Phi(\Delta t) = cos(t\mathbf{w})$.

**Batch-wise Linear MMD Approximation.** Recall that the primary goal of this module is to cultivate task-invariant representations by minimizing the distribution discrepancies across all tasks. MMD as a powerful non-parametric metric, effectively compares distributions based on two data sets. In many practical scenarios, the distribution shift within a graph is traditionally measured using the classical MMD approximation, $\text{MMD}_u^2$ (details in Appendix C) across the entire graph (Gretton et al., 2012; Zellinger et al., 2017). However, $\text{MMD}_u^2$ involves pairwise similarity calculations,

leading to quadratic time complexity. And it is unsuitable for batch-wise training due to the inconsistency of batch data sampling (Yan et al., 2017).

In contrast, our approach leverages the localized structure of the temporal WL subtree. This allows us to implement an unbiased approximation of MMD that operates with linear computational complexity (Gretton et al., 2012). Additionally, our temporal WL subtree can capture the structure information from other batches of data with its root patterns, allowing us to measure distribution shifts in batch training required by TGNNs, significantly reducing the information loss typically associated with batch data isolation (refer to the experiment in Sec. 5.4).

**Definition 3** *Linear MMD Approximation. This linear approximation assumes that the datasets being compared contain an equal number of instances $m$. The estimator is an unbiased estimate of* $\mathrm{MMD}^2[\mathcal{F}_{mmd}, p, q]$, *where $p$ and $q$ denote targets and source data distribution.*

$$\mathrm{MMD}^2_\ell[\mathcal{F}_{mmd}, \mathbb{G}_{k-1}, \mathbb{G}_k] := \frac{1}{m} \sum_{i=1}^m f_{mmd}\left((\boldsymbol{x}^{k-1}_{2i-1}, \boldsymbol{x}^{k-1}_{2i}), (\boldsymbol{x}^k_{2i-1}, \boldsymbol{x}^k_{2i})\right)$$

$$f_{mmd}\left(\boldsymbol{x}^{k-1}_{2i-1}, \boldsymbol{x}^{k-1}_{2i}), (\boldsymbol{x}^k_{2i-1}, \boldsymbol{x}^k_{2i})\right) = \psi(\boldsymbol{x}^{k-1}_{2i-1}, \boldsymbol{x}^{k-1}_{2i}) + \psi(\boldsymbol{x}^k_{2i-1}, \boldsymbol{x}^k_{2i})$$
$$-\psi(\boldsymbol{x}^{k-1}_{2i-1}, \boldsymbol{x}^k_{2i}) - \psi(\boldsymbol{x}^{k-1}_{2i}, \boldsymbol{x}^k_{2i-1})$$

The LTDO module addresses the challenge of measuring distribution shifts between adjacent tasks by focusing on local structures within each batch. By comparing batch data from successive tasks, we ensure that the data volumes align with the assumptions required for linear MMD computation. Additionally, batches are organized in chronological order to define the time domains, enabling comparisons within the same temporal contexts, such as comparing data from spring to spring, to enhance the accuracy of our discrepancy measurements. The measurement of distribution discrepancies between successive tasks is defined as follows.

$$div(G_{k-1}, G_k) = \frac{1}{|\mathcal{B}|} \sum_{i=0}^{|\mathcal{B}|} div(B^i_{k-1}, B^i_k); \ div(B^i_{k-1}, B^i_k) = \sum_{n=0}^d \mathrm{MMD}_\ell\left(C^n(B^i_{k-1}), C^n(B^i_k)\right) \tag{7}$$

Here, $|\mathcal{B}|$ denotes the number of batches in the current task, $B^i_k$ represents the $i^{th}$ batch in task $k$, and $|B|$ denotes the size of batch representing instances $m$ in the $\mathrm{MMD}^2_l$. The function $C\left(B^i_k\right) = [C^d_s(v) \| C^d_t(v), v \in B^i_k]$ represents the features of the temporal WL subtree patterns for batch data, where features of structural and temporal patterns are concatenated. The discrepancy loss function is defined as follows.

$$\mathcal{L}_{DIV} = div(G_{k-1}, G_k) \tag{8}$$

**Time Complexity and Effectiveness Analysis.** $\mathrm{MMD}^2_l$ requires only $\mathcal{O}(1)$ memory, and $\mathcal{O}(n)$ time complexity to compute the kernel on all interacting pairs. Conversely, the classical MMD approximation, $\mathrm{MMD}^2_u$, demands $\mathcal{O}(n)$ memory and $\mathcal{O}(n^2)$ computational time, making it less efficient (details in Appendix C). Given the topology of the temporal WL subtree, the time complexity of the LTDO module is $\mathcal{O}(\frac{1}{2}(|\mathcal{V}| \cdot |\tilde{S}|))$ in the worst case, where the batch size equals the size of the input graph. $|\tilde{S}|$ denotes the average size of the subtree, depending on the neighborhood size and depth. Since $|\tilde{S}|$ is a small constant, the LTDO module reduces the computational complexity of discrepancy measurement from $\mathcal{O}(|\mathcal{V}|^2)$ to a linear time complexity $\mathcal{O}(|\mathcal{V}|)$.

According to Hoeffding (1963), the deviation bound of $\mathrm{MMD}^2_l$ is $\mathrm{Pr}_{G_{k-1}, G_k}\{\mathrm{MMD}^2_l(\mathbb{G}_{k-1}, \mathbb{G}_k) -\mathrm{MMD}^2(G_{k-1}, G_k) > t\} \leq \exp\left(-\frac{t^2|B|}{4K^2}\right)$, assuming that $(0 \leq \psi(x_i, x_j) \leq K)$. This states the probability that the empirical $\mathrm{MMD}^2_l$ deviates from the squared population MMD by more than $t$, where $t$ is a predefined threshold. $\mathrm{MMD}^2_u$ has the same Hoeffding's bound as $\mathrm{MMD}^2_l$. Additionally, the asymptotic variance for $\mathrm{MMD}^2_l$ is given by $\mathrm{Var}_{z,z'}[f_{mmd}(z, z')]$ when $\mathrm{MMD}^2_l$ converges in distribution to a Gaussian. The asymptotic variance for $\mathrm{MMD}^2_u$ is the variance of $\mathbb{E}_{z'}[f_{mmd}(z, z')]$ (Gretton et al., 2012) in the same converges scenario.

The analysis demonstrates that the LTDO module significantly reduces computational complexity while maintaining comparably high accuracy relative to classical quadratic time MMD estimators. Further empirical validation of these theoretical advantages is detailed in Section 5.4.

### 4.3 N-FORGOT

The overall optimization objective of the whole framework N-FORGOT is designed as follows:

$$\mathcal{L} = \mathcal{L}_{CL} + \alpha(\mathcal{L}_{DB} + \mathcal{L}_{IC}) + \beta\mathcal{L}_{DIV} \tag{9}$$

Here the classification loss, $\mathcal{L}_{CL}$, represents the categorical cross-entropy loss. The hyperparameters $\alpha$ and $\beta$ can adjust the balance between preventing forgetting and generalization. We provide the pseudo-code of **N-ForGOT** in Appendix D.

## 5 EXPERIMENTS

In this section, we assess **N-ForGOT** in terms of its overall effectiveness, its capability to manage forgetting, and its ability to generalize unseen data distribution.

### 5.1 EXPERIMENTAL SETUP

**Datasets.** We assessed **N-ForGOT** using four real-world datasets: Yelp (Sankar et al., 2020), Reddit (Baumgartner et al., 2020), Taobao (Du et al., 2019), and Amazon (Hou et al., 2024). We summarize the statistics of datasets in Table 1. We adopt the Reddit and Taobao datasets from previous OTGL work (Feng et al., 2023) and follow their conventions for dataset construction in the other two datasets. More information about the datasets can be found in Appendix E.1.

Table 1: Dataset Statistic

|  | Yelp | Reddit | Taobao | Amazon |
|---|---|---|---|---|
| **# Nodes** | 20,992 | 10,845 | 114,232 | 361,965 |
| **# Edges** | 234,459 | 216,397 | 455,662 | 4,172,700 |
| **# Tasks** | 6 | 6 | 3 | 7 |
| **# Timespan / task** | 1 year | 1 month | 2 days | 3 months |
| **# Classes / task** | 2 | 3 | 30 | 2 |

**Evaluate Metric.** For each task, we use 80% nodes for training, 10% nodes for validation, and 10% nodes for testing. To comprehensively evaluate our method, we use three widely-used metrics (Wang et al., 2024) in continual learning to evaluate our method: Average Performance (AP), Average Forgetting (AF), and Backward Transfer (BWT). **AP** measures the mean accuracy of the model across all tasks, calculated as $AP = \frac{1}{K}\sum_{i=1}^{K} a_i$, where $a_i$ denotes the ACC (Accuracy) on the $i^{th}$ task, $K$ is the total number of tasks. **AF** quantifies the loss of previously acquired knowledge, defined by the difference between the maximum historical performance of a task and its current performance: $AF = \frac{1}{K-1}\sum_{j=1}^{K-1}\left(\max_{i\in\{1,...,K-1\}}(a_{K,j} - a_{i,j}), \forall j \le K\right.$. **BWT** reflects the average influence of learning a new K-th task on the performance of all previously learned tasks. It is defined as $BWT = \frac{1}{K-1}\sum_{j=1}^{K-1}(a_{K,j} - a_{j,j})$. This metric evaluates how the introduction of new tasks can positively impact the performance of previous tasks. These three metrics collectively provide a comprehensive evaluation of our method in not only preventing catastrophic forgetting but also enhancing generalizability across tasks.

**Baseline Methods.** We benchmark our approach against Joint training approach, Finetune training approach, and six models across both computer vision (EWC (Kirkpatrick et al., 2016), and LwF (Li & Hoiem, 2016)) and graph-structured data domains (TWP (Liu et al., 2021) and ER-GNN (Zhou & Cao, 2021), OTGNet (Feng et al., 2023)). Joint training, which trains the model simultaneously on data from all tasks, does not follow the OTGL setting. This approach provides a useful upper bound of AP. Finetune training provides a lower bound of forgetting. Details about the baseline and experiment setting can be found in Appendix E.2.

### 5.2 PERFORMANCE COMPARISON

**Performance Comparison with Baselines.** Table 2 summarizes the overall performance from ten runs. Our method closely matches the AP performance of joint training and significantly outperforms other baselines. Specifically, N-ForGOT achieves a minimum of 3.07% and up to 18.30% higher AP compared to other baselines. Furthermore, the results for metrics AF and BWT demonstrate that N-ForGOT minimizes forgetting and improves performance on previous tasks during the continual learning of task sequences, thereby demonstrating effective generalization. Key observations include:

*Managing Catastrophic Forgetting:* Our method has demonstrated exceptional AF performance across all task sequences, both long and short, achieving up to 89.73% and at least 14.34% reductions in forgetting rates compared to the closest baseline. The largest gap in AF results between N-ForGOT and the baselines occurs in the Reddit datasets. In the experiments conducted on the Reddit dataset, methods that preserve knowledge by replaying data from previous tasks in the current task, such as ER-GNN (Zhou & Cao, 2021) and OTGNet (Feng et al., 2023), demonstrate poorer ability in managing catastrophic forgetting compared to methods based on parameter regulation. This may be due to the varied data distributions, which make it challenging for these methods to sample important structures that can represent temporal topological information effectively for replay.

*Generalizability:* Higher BWT values represent positive impacts on the performance of previous tasks as new tasks are learned. Our method consistently demonstrates higher BWT values, particularly in datasets with long task sequences. This positive impact suggests our method can swiftly adapt to unseen data distributions introduced by incoming tasks. The detailed analysis of the BWT results across continually learned task sequences is introduced in Sec 5.3.

**Module Ablation.** N-ForGOT with only the TICR module (refers to N-ForGOT w/o LTDO in Table 2) achieves the lowest AF, indicating its ability to manage forgetting issues. N-ForGOT equipped solely with the LTDO module (refers to N-ForGOT w/o TICR in Table 2) shows better performance compared to when using both modules, but it incurs greater forgetting. This phenomenon is likely due to the LTDO module performing the higher performance peak of target tasks during sequential continual learning, as AF is measured by max performance. The module ablation study demonstrates that the TICR module effectively addresses forgetting, and the LTDO module enhances average performance across all tasks. Our method combines the two modules for dynamic adaptation to new tasks while preserving previous learning. An extensive analysis of two plug-in modules is conducted through a hyperparameter ablation study, which assesses the roles of hyperparameters $\alpha$ and $\beta$ as outlined in Equation 9, with details provided in Appendix E.5. The results demonstrate the crucial role of both plug-in modules in balancing the retention of previous data distributions with the adaptation to incoming data.

Table 2: Comparisons (%) analysis and module ablation study of N-ForGOT with baselines.

| Method | Yelp | | | Reddit | | | TaoBao | | | Amazon | | |
|---|---|---|---|---|---|---|---|---|---|---|---|---|
| | AP (↑) | AF (↑) | BWT (↑) | AP(↑) | AF (↑) | BWT (↑) | AP (↑) | AF (↑) | BWT (↑) | AP (↑) | AF (↑) | BWT (↑) |
| Joint | 70.855 (±1.03) | - | - | 76.455 (±3.11) | - | - | 87.910 (±1.83) | - | - | 83.753 (±0.95) | - | - |
| Finetune | 61.505 (±1.97) | -27.832 (±3.07) | -27.832 | 46.506 (±4.63) | -13.050 (±5.27) | -12.929 | 56.050 (±2.62) | -44.158 (±2.11) | -44.158 | 69.853 (±1.67) | -34.532 (±1.31) | -34.532 |
| EWC | 52.425 (±2.74) | -16.887 (±4.01) | -16.445 | 45.983 (±4.79) | -20.794 (±4.48) | -19.620 | 70.068 (±1.18) | -23.104 (±3.33) | -23.103 | 51.983 (±1.48) | -30.362 (±1.02) | -30.362 |
| LwF | 54.482 (±1.45) | -23.462 (±3.67) | -23.461 | 51.626 (±3.20) | -18.047 (±3.08) | -18.047 | 68.108 (±2.30) | -22.294 (±0.77) | -22.294 | 61.875 (±0.95) | -25.820 (±3.85) | -25.820 |
| TWP | 60.537 (±5.01) | -15.616 (±2.27) | -15.185 | 52.985 (±1.24) | -9.973 (±4.18) | -9.834 | 80.895 (±2.09) | -19.114 (±2.69) | -19.114 | 68.785 (±3.05) | -25.835 (±1.80) | -25.651 |
| ER-GNN | 63.713 (±3.24) | -14.958 (±3.55) | -14.724 | 43.756 (±4.45) | -30.252 (±2.96) | -30.252 | 72.159 (±1.99) | -14.104 (±2.29) | -14.104 | 73.941 (±1.98) | -27.071 (±1.40) | -27.070 |
| OTGNet | 61.947 (±2.45) | -4.912 (±2.07) | -3.156 | 44.930 (±4.98) | -35.540 (±5.20) | -35.540 | 77.853 (±1.07) | -28.926 (±1.11) | -28.926 | 77.535 (±2.01) | -18.095 (±0.78) | -17.623 |
| **N-ForGOT** | **69.356** (±1.07) | **-2.829** (±1.72) | **-2.050** | **62.681** (±3.85) | **-1.024** (±0.23) | **-0.889** | **83.376** (±1.55) | **-6.529** (±1.36) | **-6.529** | **80.300** (±1.98) | **-15.500** (±1.54) | **-12.965** |
| (w/o LTDO) | 65.342 (±2.63) | -1.932 (±0.90) | -1.932 | 59.710 (±4.08) | -1.021 (±0.77) | -1.021 | 76.850 (±2.68) | -5.364 (±0.47) | -5.364 | 76.924 (±1.62) | -12.318 | -12.318 |
| (w/o TICR) | 70.268 (±0.88) | -6.765 (±0.45) | -3.975 | 64.721 (±3.01) | -6.014 (±2.60) | -1.980 | 78.074 (±1.65) | -11.093 (±1.73) | -11.093 | 89.621 (±1.08) | -21.524 (±2.20) | -21.524 |

## 5.3 PERFORMANCE ANALYSIS ACROSS TASKS

We further analyze the effectiveness of our model in mitigating catastrophic forgetting and enhancing generalization ability with the increased tasks.

**Performance Trends Across Tasks.** We plot the trends of AP values in Figure 3, illustrating the performance trajectory throughout the continual learning process. The performance curve of our method, represented by the red line, demonstrates increased performance compared to those of other baselines. This indicates that N-ForGOT not only addresses the forgetting issue but

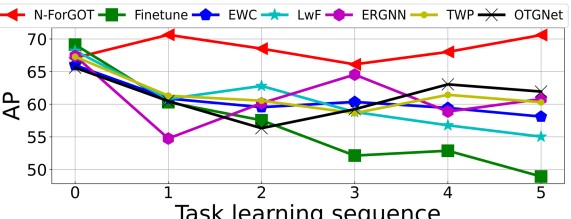

Figure 3: Performance trends throughout the continual learning process.

also improves task performance by enhancing generalizability across data distributions through task discrepancy optimization. Specifically, the increased performance observed upon the introduction of Task 1 and Task 4 suggests that our method effectively adapts to newly arriving tasks.

**BWT Trends Across Tasks.** We analyze performance changes for the target task with new tasks continually introduced, as shown in Figure 5. The x-axis represents the 'Target Task', and the y-axis shows the new tasks introduced continually. This heatmap effectively highlights how each new

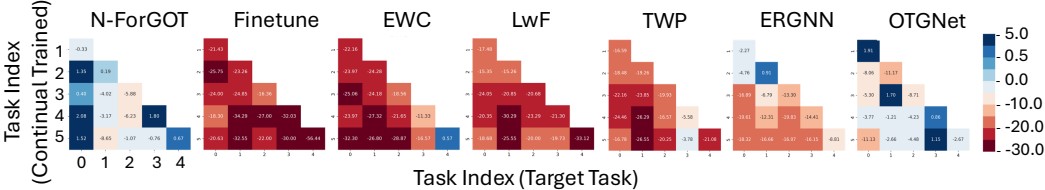

Figure 4: BWT trends across tasks. Blue cells indicate performance improvements in the target task during the continual learning of task sequences. Conversely, cells in red denote performance drop.

task impacts the performance of previously learned tasks. Each cell in the heatmap indicates the impact on the performance of the target task following the introduction of new tasks, as measured by the BWT metrics. Cells colored blue signify improved performance, indicating that new tasks have positively influenced the target task's performance. Conversely, cells in red denote significant forgetting. Notably, for Task 0, our method demonstrates significant performance improvement after successively learning the next five tasks (Task 1-5). Our method not only minimizes forgetting but also occasionally shows positive backward transfer, where the introduction of new tasks enhances the performance of earlier tasks.

### 5.4 DISCREPANCY MEASUREMENT STRATEGY ANALYSIS

We conduct further analysis of our proposed discrepancy measurement strategy, the LTDO module, focusing on both its efficiency and effectiveness.

**Efficiency Analysis.** As shown in Figure 5 (left), we compare the efficiency of our discrepancy measurement method (LTDO module) with the classical MMD measurement. For a fair comparison, we use a single Gaussian kernel as the kernel mapping function for both methods due to the classical MMD's inability to

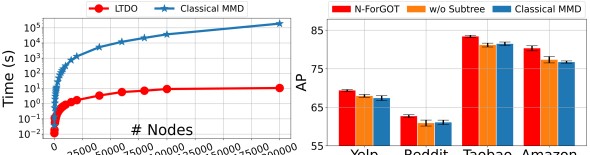

Figure 5: Discrepancy measurement strategy analysis: efficiency analysis (left), performance analysis (right).

support multiple kernels within our device's memory limitations. Further details on kernel design in the main experiment are available in Appendix E.2. We observe that the LTDO module becomes significantly more efficient than classical MMD as the size of the graph increases. This comparison demonstrates the efficiency gains achieved through our proposed discrepancy measurement method in graph data scenarios.

**Performance Analysis.** We analyze the effectiveness of our temporal WL subtree and explore the performance differences between classical and linear MMD approaches, as illustrated in Figure 5 (right). The term 'w/o Subtree' denotes that our method measures the discrepancy between tasks directly using node features, without involving subtree structures. Excluding temporal WL subtree patterns significantly reduces performance, demonstrating their effectiveness in capturing both temporal and structure distribution shifts and mitigating accuracy loss due to batch data isolation. Additionally, the term 'Classical MMD' refers to our experiment using classical MMD as opposed to linear MMD for discrepancy measurement. Despite theoretical analyses suggesting that classical MMD offers greater stability (as shown in Sec.4.2), our experimental results indicate that our method does not experience a performance drop when enhancing efficiency.

We also conduct sensitivity analysis of hyperparameters which can be found in Appendix E.4.

### 6 CONCLUSION

This study introduces **N-ForGOT**, a novel approach to Open Temporal Graph Learning (OTGL) that tackles both forgetting and distribution shifts. To our knowledge, this is the first work to extend forgetting mitigation in OTGL by incorporating a theoretical analysis. Our method integrates two plug-in modules for Temporal Graph Neural Networks (TGNNs): one that preserves previously learned knowledge through class decision boundaries and inter-class connections, and another that ensures broad generalization across both historical and emerging data distributions, utilizing an effective linear Maximum Mean Discrepancy approximation with a novel temporal Weisfeiler-Lehman subtree representation for measuring distribution shifts. Experiments on four real-world datasets demonstrate that **N-ForGOT** not only minimizes forgetting but also enhances performance on previous tasks, showcasing its effectiveness and generalizability. Future work will focus on extending our framework to handle unseen graph data from diverse domains and developing a model capable of generating varied graph structures.

ACKNOWLEDGMENTS

Lei Chen's work is partially supported by National Key Research and Development Program of China Grant No. 2023YFF0725100, National Science Foundation of China (NSFC) under Grant No. U22B2060, Guangdong-Hong Kong Technology Innovation Joint Funding Scheme Project No. 2024A0505040012, the Hong Kong RGC GRF Project 16213620, RIF Project R6020-19, AOE Project AoE/E-603/18, Theme-based project TRS T41-603/20R, CRF Project C2004-21G, Guangdong Province Science and Technology Plan Project 2023A0505030011, Guangzhou municipality big data intelligence key lab, 2023A03J0012, Hong Kong ITC ITF grants MHX/078/21 and PRP/004/22FX, Zhujiang scholar program 2021JC02X170, Microsoft Research Asia Collaborative Research Grant, HKUST-Webank joint research lab and 2023 HKUST Shenzhen-Hong Kong Collaborative Innovation Institute Green Sustainability Special Fund, from Shui On Xintiandi and the InnoSpace GBA.

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

# A  PROOF OF THE THEOREM 1

Based on the preliminary definition provided in Sec 3.1, we introduce the detailed definition of the population risk in this case.

**Definition 4** *For a hypothesis function $f$, the probability according to the data distribution that a hypothesis disagrees with the labeling function $\hat{f}$, is defined as:*

$$\mathcal{R}_{\mathbb{G}}(f, \hat{f}) = \mathrm{E}_{x \sim \mathbb{G}}[|f(x) - \hat{f}(x))|]$$

*We use the shorthand $\mathcal{R}_{\mathbb{G}}(f) = \mathcal{R}_{\mathbb{G}}(f, \hat{f})$ to refer to the risk of a hypothesis [3].*

Our method aims to minimize the population risk of the current task, $\mathcal{R}_k(f)$ [4]. and reduce the difference between the current and previous task risks, $|\mathcal{R}_k(f) - \mathcal{R}_{1:k-1}(f)|$. Theorem 1 is derived by first assessing $|\mathcal{R}_k(f) - \mathcal{R}_{1:k-1}(f)|$. We have:

$$
\begin{aligned}
&|\mathcal{R}_k(f) - \mathcal{R}_{1:k-1}(f)| \\
= &|\mathcal{R}_k(f) - \mathcal{R}_{1:k-1}(f) + \mathcal{R}_{1:k-1}(f, \hat{f}_k) - \mathcal{R}_{1:k-1}(f, \hat{f}_k)|
\end{aligned}
\tag{10}
$$

Based on the triangle inequality (Def A.1 in (Mohri et al., 2018)) for classification error, which implies that for any labeling functions $f_1$, $f_2$ and $f_3$, we have $\mathcal{R}(f_1, f_2) \leq \mathcal{R}(f_2, f_3) + \mathcal{R}(f_2, f_3)$. Thus, we can reorganize Equation 11 as follows:

$$
\begin{aligned}
\mathcal{R}_k(f) \leq\ & \mathcal{R}_{1:k-1}(f) + |\mathcal{R}_{1:k-1}(f, \hat{f}_k) - \mathcal{R}_{1:k-1}(f, \hat{f}_{1:k-1})| \\
& + |\mathcal{R}_k(f, \hat{f}_k) - \mathcal{R}_{k-1}(f, \hat{f}_k)| \\
\leq\ & \mathcal{R}_{1:k-1}(f) + \mathbb{E}_{1:k-1}[|\hat{f}_{1:k-1}(x) - \hat{f}_k(x)|] \\
& + |\mathcal{R}_k(f, \hat{f}_k) - \mathcal{R}_{1:k-1}(f, \hat{f}_k)| \quad \textbf{(Definition 4)} \\
\leq\ & \mathcal{R}_{1:k-1}(f) + \mathbb{E}_{1:k-1}[|\hat{f}_{1:k-1}(x) - \hat{f}_k(x)|] \\
& + div_{\mathcal{H}}(\mathbb{G}_{1:k-1}, \mathbb{G}_k) \quad \textbf{(Lemma 1)} \\
\leq\ & \hat{\mathcal{R}}_{1:k-1}(f) + div_{\mathcal{H}}(\mathbb{G}_{1:k-1}, \mathbb{G}_k) + \mathbb{E}_{1:k-1}[|\hat{f}_{1:k-1}(x) - \hat{f}_k(x)|] \\
& + \sqrt{\frac{2d}{m}\log(\frac{em}{d})} + \sqrt{\frac{1}{2m}\log\left(\frac{1}{\delta}\right)} \quad \textbf{(Lemma 2)}
\end{aligned}
$$

We could alternatively add $\mathcal{R}1(f, \hat{f}k)$ instead of $\mathcal{R}k(f, \hat{f}1)$ in the first line. This would yield the same bound but with the expectation taken over $\mathbb{G}1$ instead of $\mathbb{G}k$. The smaller bound holds, leading to:

$$
\begin{aligned}
\mathcal{R}_k(f) \leq\ & \hat{\mathcal{R}}_{k-1}(f) + div_{\mathcal{H}}(\mathbb{G}_{1:k-1}, \mathbb{G}_k) \\
& + \sqrt{\frac{2d}{m}\log(\frac{em}{d})} + \sqrt{\frac{1}{2m}\log\left(\frac{1}{\delta}\right)} \quad + \xi
\end{aligned}
$$

where $\xi = \min\left\{\mathbb{E}_k\left[|f_{1:k-1}(\mathbf{x}) - f_k(\mathbf{x})|\right], \mathbb{E}_{1:k-1}\left[|f_{1:k-1}(\mathbf{x}) - f_k(\mathbf{x})|\right]\right\}$

**Lemma 1** *For the definition of hypothesis space $\mathcal{H}$ and the symmetric difference hypothesis space $\mathcal{H}\Delta\mathcal{H}$ (Ben-David et al., 2010),*

$$
\begin{aligned}
&|\mathcal{R}_k(f, \hat{f}_k) - \mathcal{R}_{k-1}(f, \hat{f}_k)| \\
= &\mathrm{E}_k[|f(x) - \hat{f}(x))|] - \mathrm{E}_{1:k-1}[|f(x) - \hat{f}(x))|] \\
\leq &\sup_{\|f\|_{\mathcal{H}} \leq 1} \mathbb{E}_k[f(x)] - \mathbb{E}_{1:k-1}[f(x)] \\
\leq &\ 1/2\ div_{\mathcal{H}}(\mathbb{G}_{1:k-1}, \mathbb{G}_k)
\end{aligned}
$$

---

[3] Here, $\mathcal{R}_{\mathbb{G}_k}(f)$ is equivalent to $\mathcal{R}_{\mathbb{G}_k}$ as defined in Section 3.

[4] For simplicity, we use $\mathcal{R}_k$ to represent $\mathcal{R}_{\mathbb{G}_k}$ in the following proof.

**Lemma 2** *According to the Theorem 11.8 in (Mohri et al., 2018). Let $\mathcal{F}$ be a family of real-valued functions and the family of loss functions associated with $\mathcal{F}$ is non-negative. Assume $Pdim(\mathcal{F}) = d$. Then $\forall \delta > 0$, with probability at least $1 - \delta$ over the choice of a sample size $m$ and natural exponential $e$, the following inequality holds:*

$$\mathcal{R}(f) \leq \hat{\mathcal{R}}(f) + \sqrt{\frac{2d}{m} \log(\frac{em}{d})} + \sqrt{\frac{1}{2m} \log\left(\frac{1}{\delta}\right)}$$

## B  THEORTICAL FOUNDATION FOR THE DEFINITION 2

The theoretical foundation for Definition 2 is established by the principles of the WL algorithm (Leman & Weisfeiler, 1968) and the nature of TGNNs, as applied to the isomorphism problem in temporal graphs. Here is the proposed Proposition and its proof:

**Theorem 2** *Let $G_k$ and $G_{k-1}$ be any two non-isomorphic temporal graphs. If a neighborhood-based massage passing TGNNs obtains different multisets of node embedding for $G_k$ and $G_{k-1}$. Then, the temporal WL subtree algorithm determines that $G_k$ and $G_{k-1}$ are not isomorphic.*

*Proof.* Based on the foundational theory (Morris et al., 2019), we consider any two nodes from a temporal graph $\mathbf{G} = (V, T, \mathcal{E})$. If the temporal WL returns $C^d(u) = C^d(v)$ for nodes $u$ and $v$, it implies that their corresponding embeddings from the TGNN are identical, i.e., $g^d(u) = g^d(v)$. This assertion can be proved through induction.

$d = 0$: Initially, the temporal WL algorithm uses the initial node features as colors, similar to how TGNNs use these features as embeddings. Thus, if $C_s^0(u) = C_s^0(v)$, then $g^0(u) = g^0(v)$ holds trivially.

$d > 0$: Assume the proposition holds for depth $d$. The color of a node v, $C_s^d$, output from the WL algorithm at iteration $d$ depends on the multiset of colors at iteration $d-1$ of its neighbors. Applying the induction hypothesis, for any two nodes $u, v$, such that $C_s^{d+1}(u) = C_s^{d+1}(v)$, it follows:

$$\begin{aligned} &\left(C_s^d(u), \left\{\left\{(C_s^d(i), e_{iu}(t), t) : (u, i, t) \in \mathcal{E}\right\}\right\}\right) \\ =& \left(C_s^d(v), \left\{\left\{(C_s^d(j), e_{jv}(t), t) : (v, j, t) \in \mathcal{E}\right\}\right\}\right) \end{aligned} \tag{11}$$

This equivalence translates to the TGNN outputs:

$$\begin{aligned} &\left(g^d(u), \left\{\left\{(g^d(i), e_{iu}(t), t) : (u, i, t) \in \mathcal{E}\right\}\right\}\right) \\ =& \left(g^d(v), \left\{\left\{(g^d(j), e_{jv}(t), t) : (v, j, t) \in \mathcal{E}\right\}\right\}\right) \end{aligned} \tag{12}$$

From this, we conclude that if the multisets of temporal WL subtree patterns $\{\{C^d(u)\}\}_{u \in V} = \{\{C^d(v)\}\}_{v \in V}$, then $\{\{g^d(u)\}\}_{u \in V} = \{\{g^d(v)\}\}_{v \in V}$. Hence, a neighborhood-based massage passing TGNN with $d - layer$ will output the same result for nodes with the same depth in the temporal WL subtree pattern.

## C  GENERAL MMD

**Definition 5** *Maximum Mean Discrepancy*. *Given $x^k$ independent random variables with distribution $\mathbb{G}_k$, and $x^{k-1}$ independent random variables with distribution $\mathbb{G}_{k-1}$. An unbiased empirical estimate of MMD (Gretton et al., 2012) can be defined as,*

$$\mathrm{MMD}_u^2[\mathbb{G}_{k-1}, \mathbb{G}_k] = \frac{1}{m(m-1)} \sum_{i=1}^{m} \sum_{j \neq i}^{m} \psi\left(x_i^{k-1}, x_j^{k-1}\right)$$

$$+ \frac{1}{n(n-1)} \sum_{i=1}^{n} \sum_{j \neq i}^{n} \psi\left(x_i^k, x_j^k\right) - \frac{2}{mn} \sum_{i=1}^{m} \sum_{j=1}^{n} \psi\left(x_i^{k-1}, x_j^k\right)$$

## D  TRAINING PROCEDURE

We provide the pseudo-code for our training procedure in Algorithm 1.

---

**Algorithm 1** Continual Learning Algorithm

---
1: **Input:** Continual tasks $\mathcal{K} = \{\mathcal{K}_1, \ldots \mathcal{K}_K\}$,
2: **Output:** Optimized model $f(\theta^*)$, Predicted label $\mathbf{Y}_{1:k}$,
3: Initialize $\theta$ randomly;
4: **for** $\mathbf{G}_k, Y_k \in \mathcal{K}_K$ **do**
5:      Split the training data into batches $\mathbf{B}_k$ in chronological orders
6:      **for** $B_k^i$ in $\mathbf{B}_k$ **do**
7:          $\hat{y}_k', \hat{y}_k \leftarrow f(\theta_{k-1}, B_k^i), f(\theta_k, B_k^i)$
8:          $\mathcal{L}_{DB} \leftarrow \hat{y}_k', \hat{y}_k$          $\triangleright$ Eq. 3
9:          $\hat{p}_k, \hat{p}_k \leftarrow g_{k-1}(\theta_{k-1}, B_k^i), g_k(\theta_k, B_k^i)$          $\triangleright$ Eq. 6
10:         $\mathcal{L}_{IC} \leftarrow \hat{p}_k, \hat{p}_k$          $\triangleright$ Eq. 5
11:         Construct temporal WL subtree set $S_k^i \leftarrow B_k^i$          $\triangleright$ Prop. 2
12:         Obtain features of subtree patters $C^n(v|B_k^i)$          $\triangleright$ Eq. 8,9
13:         $\mathcal{L}_{DIV} = \mathrm{div}(B_{k-1}^i, y_k^i)$          $\triangleright$ Eq. 12
14:         $\mathcal{L}_{classification} \leftarrow \mathrm{CrossEntropy}\,(y, \hat{y})$
15:         $\theta \leftarrow \arg\min(\mathcal{L}_{classification} + \alpha(\mathcal{L}_{DB} + \mathcal{L}_{IC}) + \beta\mathcal{L}_{DIV})$
16:      **end for**
17:      $\mathbf{P}_k \leftarrow (g_k(\theta_k^*, G_k), \mathbf{P}_{k-1})$          $\triangleright$ Eq. 4
18: **end for**

---

## E  EXPERIMENT

### E.1  DETAILS OF DATASETS

Specifically, in the Yelp dataset, businesses are represented as nodes. A temporal edge is established between two businesses if a user comments on both within a one-week interval. We organize the dataset chronologically, treating data from each year as a distinct task. To simulate the introduction of new classes in OTG, we selectively sample two large sets of business categories that have not appeared in previous tasks, treating these as new classes. In the Reddit dataset, posts are treated as nodes and the community topic of the post is the node label. Connections between posts are established in the same manner as in the Yelp dataset. For the Taobao and Amazon datasets, products are represented as nodes, and a product-to-product graph is constructed using the same criteria as for Yelp.

Table 3: Yelp.

|  | Task 0 | Task 1 | Task 2 | Task 3 | Task 4 | Task 5 |
|---|---|---|---|---|---|---|
| **# Nodes** | 1905 | 1621 | 1488 | 2617 | 1088 | 12273 |
| **# Edges** | 15702 | 7352 | 2995 | 4383 | 2351 | 201676 |

Table 4: Reddit.

|  | Task 0 | Task 1 | Task 2 | Task 3 | Task 4 | Task 5 |
|---|---|---|---|---|---|---|
| **# Nodes** | 1706 | 1749 | 1853 | 1677 | 1691 | 2338 |
| **# Edges** | 12558 | 25610 | 23573 | 24950 | 61702 | 68004 |

### E.2  EXPERIMENT SETTING

We assess our proposed method, in a scenario where the model must differentiate among all classes from the current and previous tasks without task indicators. For example, if the model is sequentially

Table 5: Taobao.

|  | Task 0 | Task 1 | Task 2 |
|---|---|---|---|
| **# Nodes** | 15679 | 8318 | 91415 |
| **# Edges** | 59523 | 25865 | 370274 |

Table 6: Amazon.

|  | Task 0 | Task 1 | Task 2 | Task 3 | Task 4 | Task 5 | Task 6 |
|---|---|---|---|---|---|---|---|
| **# Nodes** | 9097 | 12225 | 28022 | 39906 | 58930 | 79743 | 134042 |
| **# Edges** | 31907 | 63854 | 86814 | 132214 | 330134 | 1809135 | 1718641 |

trained on two tasks with class pairs $\{(0,1),(2,3)\}$, the model is expected to classify any of the four class labels $\{0,1,2,3\}$ collectively after training on the second task. The classifications are not confined to separate task-specific pairs $\{(0,1)\}$ or $\{(2,3)\}$. This setup can demonstrate the model's adaptability across an expanding class set.

The model framework consists of three parts: a feature extractor (made up of TGNN layers), a discriminator (made up of MLP layers), and our proposed plug-in modules. For the implementation detail, we utilize a multi-layer TGNN, where each layer contains a specified number of neurons in the hidden layer. We can apply any neighborhood-based message passing TGNNs as the feature extractor. In this experiment, we use the widely used TGNNs, TGAT (Xu et al., 2020) for the feature extractor. We sample 5 neighbor nodes (from the current task) for each node to aggregate the neighborhood information. The input features for all datasets are processed in 300 dimensions. The network is trained using the Adam optimizer for each dataset with learning rates set at $\eta = 0.0001$ until convergence is reached. The classification is handled by a two-layer MLP: the first layer uses ReLU activation and Dropout, with a hidden size of 100, to transform node features into hidden representations. The second layer consists of multiple independent linear classification heads, each designed for a specific task. The entire network is optimized using cross-entropy loss to ensure effective classification performance across tasks. After training, we assess the model's performance on the current task as well as on all previous tasks. Our experiments are conducted on a GeForce RTX 3090 GPU.

For the MMD measurement, we apply the widely used Gaussian kernel as the kernel mapping function, defined as $\psi(x_i, x_j) = e^{-\|x_i - x_j\|^{2/\gamma}}$, where the bandwidth $\gamma$ is set to the median of pairwise distances among the data points. To capture the characteristics of the data at different scales, we utilize a family of m Gaussian kernels $\{\psi_u\}_{u=1}^m$, varying $\gamma_u$ based on the distance of input data and the number of kernels. We opt for five kernels to ensure a comprehensive analysis.

### E.3 BASELINES

For computer vision continual learning methods, we adapt three well-known algorithms (EWC (Kirkpatrick et al., 2016), and LwF (Li & Hoiem, 2016)) to a TGNN backbone. These methods primarily prevent CF through regularization and knowledge distillation. In the graph data domain, we compare our method with two state-of-the-art graph continual learning approaches: TWP (Liu et al., 2021) and ER-GNN (Zhou & Cao, 2021), which focus on weight preservation and dynamic regularization respectively. To ensure a fair comparison, we utilize the same TGNN as the backbone for these approaches, adapting them from their original implementations with static GNNs. Additionally, we include OTGNet (Feng et al., 2023), a replay-based class-incremental learning method specifically designed for OTGL. For fairness in our comparisons, we standardized the size of the candidate set and the replay buffer for the replay-based method to match the training durations of our method.

### E.4  Sensitivity Analysis of Hyperparameters

We analyze the effects of batch size $|B|$ and temporal WL subtree depth $\mathbf{d}$. As shown in Definition 2, we use the backbone model's number of layers to represent subtree depth. Figure 6 shows that our method is not sensitive to batch size. Figure 7 indicates that a single layer is sufficient for capturing the necessary information, demonstrating that our temporal WL subtree effectively captures sufficient details by 0-th and 1-th patterns.

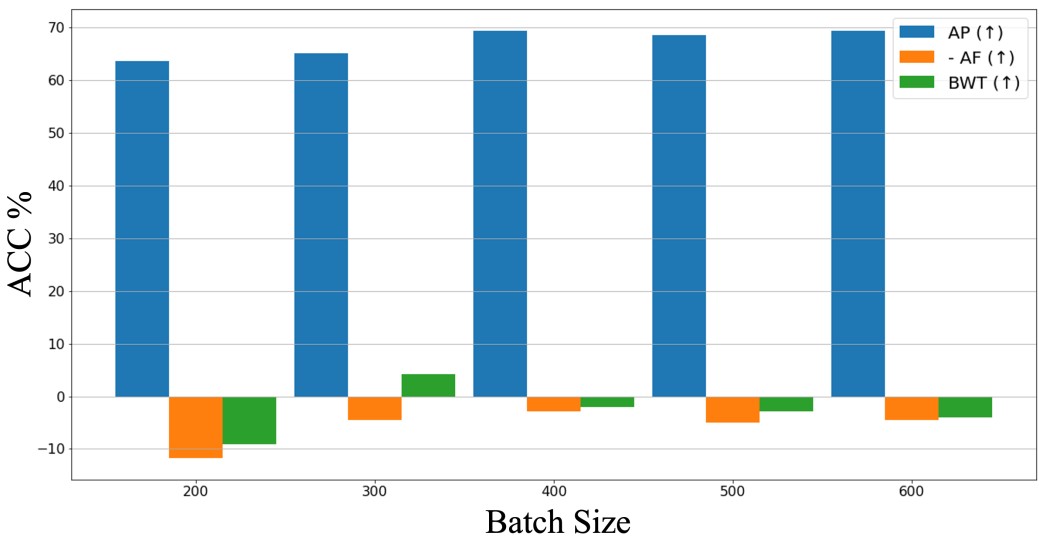

Figure 6: Sensitivity Analysis of the Batch Size

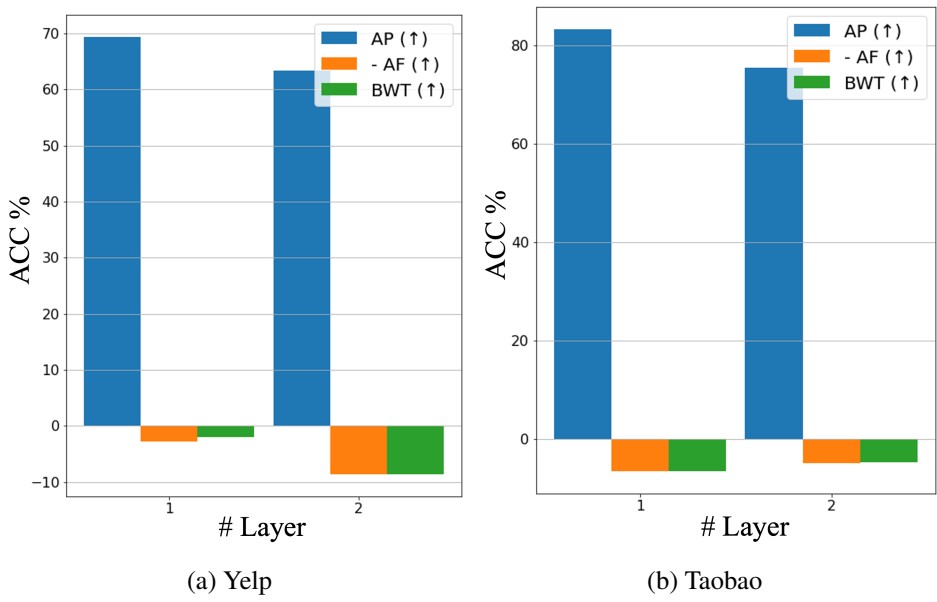

(a) Yelp                  (b) Taobao

Figure 7: Sensitivity Analysis of the Number of Layers

### E.5  Hyperparameters Study

To provide clear insights into the impact of these hyperparameters, we conducted an ablation study, with the results detailed in Table 7. The findings indicate that both plug-in modules significantly

enhance performance. The model achieves greater robustness when we assign relatively similar weights to each plug-in module in the calculation of the final loss function.

Table 7: Hyperparameters Ablation Study

| $\alpha = 1$ | $\beta = 1$ | | | $\beta = 0.8$ | | | $\beta = 0.4$ | | | $\beta = 0$ | | |
|---|---|---|---|---|---|---|---|---|---|---|---|---|
| | AP | AF | BWT | AP | AF | BWT | AP | AF | BWT | AP | AF | BWT |
| | 69.356 | -2.829 | -2.050 | 67.807 | -6.441 | -5.979 | 60.848 | -12.971 | -8.056 | 65.342 | -1.932 | -1.932 |

| $\beta = 1$ | $\alpha = 1$ | | | $\alpha = 0.8$ | | | $\alpha = 0.4$ | | | $\alpha = 0$ | | |
|---|---|---|---|---|---|---|---|---|---|---|---|---|
| | AP | AF | BWT | AP | AF | BWT | AP | AF | BWT | AP | AF | BWT |
| | 69.356 | -2.829 | -2.050 | 66.890 | -2.721 | -2.158 | 65.382 | -9.236 | -8.100 | 70.268 | -6.765 | -3.975 |

# F  TASK-TO-TASK DISTRIBUTION SHIFTS

We calculate and present the task-to-task distributional shifts, computed by MMD, in the following Table. The data presented in the following table and the heatmap (Figure 1(b)) clearly illustrate the inherent challenges posed by structural and temporal shifts in open temporal graph learning. These visualizations show how these shifts manifest across tasks, with MMD values highlighting the extent of distributional changes as new tasks are introduced. In reference to the ablation study shown in Table 2, we found that the LTDO module significantly enhances the model's effectiveness and robustness across four datasets, particularly in the context of the distribution shifts noted.

Table 8: Task-to-task distributional shifts (measured by MMD)

| Tasks | 0-1 | 1-2 | 2-3 | 3-4 | 4-5 | 5-6 |
|---|---|---|---|---|---|---|
| **Taobao Dataset** | 4.245 | 2.228 | | | | |
| **Yelp Dataset** | 1.015 | 0.681 | 0.802 | 1.068 | 1.348 | |
| **Reddit Dataset** | 1.739 | 1.456 | 2.113 | 1.921 | 1.763 | |
| **Amazon Dataset** | 2.148 | 2.005 | 1.833 | 1.785 | 1.698 | 2.790 |

