# OpenReview forum: "N-ForGOT: Towards Not-forgetting and Generalization of Open Temporal Graph Learning"
_ICLR.cc/2025/Conference — ICLR 2025 Poster_

### Official Review · Reviewer_sieS · 2024-10-27

**Soundness:** 3
**Presentation:** 2
**Contribution:** 3
**Rating:** 6
**Confidence:** 4

**Summary:**

This paper focuses on temporal graph learning and proposes a method, named N-ForGOT, to address evolution in node classes and dynamic data distributions triggered by the continuous emergence of new class labels. Two specific modules are designed to address the corresponding problem. Experimental results on four public datasets demonstrate the effectiveness of the proposed method.

**Strengths:**

The studied problem is practical and interesting.

The authors provide a theoretical analysis of the generalization error bound.

The experimental results are promising.

**Weaknesses:**

The related works are not comprehensive.

Some symbols and statements are not clear.

**Questions:**

1.	Some recent works on graph continual learning are not included in the paper, such as [1][3].
2.	The authors are encouraged to discuss the differences between the studied problem and graph continual learning more clearly as the distribution shift and forgetting problem also exist in graph continual learning [2].
3.	More recent graph continual learning baselines are encouraged to be included for comparison,
4.	When learning a new task, is the data of previous tasks available and included for training? In graph continual learning, there is no access to data from previous tasks and classes.
5.	In line 243, should the category “n” be “i”?
6.	In line 349, there are some overlapping.
7.	Please revise line 420 to make it more clear.
8.	In 484, the figure should be Figure 4 and there should be margins between the caption of Figure 4 and the main text.
9.	Why are there learning rates for the model optimization in line 892?

[1] Niu C, Pang G, Chen L. Graph Continual Learning with Debiased Lossless Memory Replay[J]. arXiv preprint arXiv:2404.10984, 2024.
[2] Zhang, X., Song, D., & Tao, D. (2022). Cglb: Benchmark tasks for continual graph learning. Advances in Neural Information Processing Systems, 35, 13006-13021.
[3] Liu, Y., Qiu, R., & Huang, Z. (2023, December). Cat: Balanced continual graph learning with graph condensation. In 2023 IEEE International Conference on Data Mining (ICDM) (pp. 1157-1162). IEEE.

---

> ### Author Response · Authors · 2024-11-23
>
> Thank you for your insightful comments. We have carefully revised the manuscript to address the concerns you raised regarding unclear writing in questions 5, 6, 7, 8, and 9. We appreciate your feedback as it has significantly contributed to improving the clarity and quality of our paper.
>
> **Comment 1:** Thanks for your mention, we have included these two papers in the related work section. These works innovatively employ condensed graphs as replay data to prevent forgetting. The condensed graphs are significantly smaller than the original, yet they consist of synthetic node representations that preserve the complete informational content of the original structures.
>
> **Comment 2:** Thank you for your suggestion. Open Temporal Graph Learning (OTGL) indeed stems from Graph Continual Learning (GCL) but is specifically tailored to address the unique challenges associated with temporal graphs. We have updated the 'Related Work' section of our manuscript to clearly delineate this.
>
> Specifically, in OTGL, the issues of distribution shift and catastrophic forgetting are more complex and pronounced compared to traditional GCL due to the dynamic nature of temporal graphs. Specifically, temporal inconsistencies result in class characteristics that evolve over time, significantly complicating the preservation of previously learned information and intensifying the problem of catastrophic forgetting. Moreover, OTGL must address distribution shifts across multiple domains, a challenge that is uniquely demanding compared to the more static scenarios encountered in GCL.  These nuances have been detailed in the Introduction section of our manuscript.
>
> **Comment 3:** Thank you for your suggestion. We acknowledge the importance of including more recent continual graph learning methods for comparison. Accordingly, we will consider incorporating the newly published works, DeLoMe [1] and CaT [2], into our baseline comparisons to provide a more comprehensive analysis. We will update the experimental results accordingly in the final version of the manuscript. Notably, DeLoMe [1], as the latest published paper, introduces a novel lossless memory replay approach that captures the holistic information of entire graphs. This method distinctively utilizes gradient-based graph condensation methods in continual graph learning, moving beyond traditional data sampling techniques to preserve more comprehensive knowledge from previous data. This innovative strategy represents a significant advancement in addressing the challenges of catastrophic forgetting in graph neural networks.
>
> **Comment 4:** Graph continual learning aims to continuously learn from new tasks in a sequence without needing to retrain all previous data. The goal is to maintain overall performance by integrating new knowledge while minimizing the forgetting of previously acquired knowledge. In this field, there is no strict prohibition against accessing data from previous tasks and classes. In fact, several studies [3][4] employ replay-based methods in graph continual learning, which selectively retrain data from previous tasks during the training of current tasks.
>
>
> [1] Niu C, Pang G, Chen L. Graph Continual Learning with Debiased Lossless Memory Replay[J]. arXiv preprint arXiv:2404.10984, 2024.
>
> [2] Liu, Y., Qiu, R., & Huang, Z. (2023, December). Cat: Balanced continual graph learning with graph condensation. In 2023 IEEE International Conference on Data Mining (ICDM) (pp. 1157-1162). IEEE.
>
> [3] Kaituo Feng, Changsheng Li, Xiaolu Zhang, and Jun Zhou. Towards open temporal graph neural
> networks. In The Eleventh International Conference on Learning Representations, ICLR, 2023.
>
> [4] Tian, Z., Zhang, D., & Dai, H. N. (2024). Continual Learning on Graphs: A Survey. arXiv preprint arXiv:2402.06330.

---

> > ### Comment · Reviewer_sieS · 2024-11-24
> >
> > I appreciate the clarification, which has addressed most of my concerns. I will maintain my positive score.

---

> > > ### Author Response · Authors · 2024-11-25
> > >
> > > Thank you for your thorough and thoughtful review. I appreciate your patience and the valuable contributions you have made to our paper.

---

### Official Review · Reviewer_Jw9d · 2024-11-04

**Soundness:** 3
**Presentation:** 3
**Contribution:** 3
**Rating:** 6
**Confidence:** 2

**Summary:**

The paper proposes N-ForGOT, a framework for Temporal Graph Neural Networks (TGNNs) that addresses the open-set problem in dynamic graphs by preserving previously learned classes while adapting to new ones. This is achieved with two main modules: the Temporal Inter-Class Connectivity Regularization (TICR) for mitigating forgetting of prior knowledge, and the Localized Temporal Graph Discrepancy Optimization (LTDO) to manage distribution shifts across tasks. Empirical results show that N-ForGOT outperforms existing methods in prediction accuracy, minimizing forgetting, and enhancing generalizability across tasks.

**Strengths:**

S1: Clarify: The overall demonstration of open-set learning in dynamic graphs is very clear, in terms of two core challenges of distribution shifts and knowledge forgetting, also with clear motivation for the proposed continual learning mechanism.

S2: Method: Two modules, TICR module and LTDO module, help the model retain essential inter-class information, reduce forgetting and preserve learned knowledge over time, and manage distribution shifts across evolving tasks, sounds rational to me.

S3: Empirical Experiments & Good Structure: The method demonstrates effective performance on multiple benchmarks, with good logic and organized well.

**Weaknesses:**

See questions.

**Questions:**

I do not have extensive expertise in this research area, but I have some general questions:

Q1: My understanding of the open-set setting is that it allows predictions on completely unseen classes. However, in the proposed method, it seems the model gradually learns across tasks by incorporating new classes over time. If the dataset consists of six tasks, does this mean the prediction space is limited to these six tasks and wouldn’t cover unknown classes beyond them? In the test set, are predictions also evaluated solely on the six classes seen during training? In this case, can we take it as the 'open-set'?

Q2: The paper mentions dynamic graph distribution shifts, but does this refer only to shifts in class labels over time? Are other types of distribution shifts, such as changes in features or interactions over time, also considered or defined in this work? And how these distribution shift issues are modeled and integrated into the proposed method?

---

> ### Author Response · Authors · 2024-11-23
>
> **Comment 1:** Thank you for your question. Our work adopts the open temporal graph learning framework as defined in "Towards Open Temporal Graph Neural Networks (OTGNet)" [1], a method highlighted among the top 5\% at ICLR 2023. This formulation is grounded in continual graph learning principles [2], where each task incorporates 'unseen classes' that are new relative to previous tasks, thereby expanding the model's predictive capacity over time. However, this setting is distinct from the 'open-set' scenario in the Open-Set Recognition problem [3], where the model encounters classes at test time that were completely unseen during training
>
> As illustrated in Figure 1(a), the exploration of temporal graphs progressively introduces new classes over time.  These newly emerging classes, compared to the historical data, expand the prediction space and challenge the model to adapt continuously. For example, according to Figure 1(a), consider a scenario where Task 1 involves graph $\\textbf{G}_ 1$ with a corresponding label set $\\mathbf{Y}_ 1$, and Task 2 introduces a graph $\\textbf{G}_ 2$ with label set $\\mathbf{Y}_ 2$. The label sets are disjoint and $\\textbf{G}_ 2$ does not overlap $\\textbf{G}_ 1$. When task 2 arrives, the model is trained on the new graph $\\textbf{G}_ 2$, but it is also expected to maintain its ability to predict across the combined label sets of both tasks: $\\mathbf{Y}_ 1 \\cup  \\mathbf{Y}_ 2$.
>
> The main challenge of open temporal graph learning is to prevent catastrophic forgetting while continuously learning from new tasks in a sequence without needing to retrain previous data. Beyond this, our approach aims to balance knowledge retention from past data with effective adaptation to evolving data distributions. Based on the results theoretically analyzed in Theorem 1, we have proposed two plug-in modules specifically designed to address both forgetting and distribution shift problems.
>
> **Comment 2:**
> Thank you for your thoughtful question.  The dynamic graph distribution shifts mentioned in our paper encompass more than just shifts in class labels over time; they also include changes in node features, interaction frequencies, and overall graph structure.
>
> In this work, we quantify the distribution shifts in open temporal graphs through two primary types: structure shift and temporal shift (as mentioned in lines 068-070).
> **Structure shifts** arise from differences in graph structures between old and new data, such as the emergence of new interactions or changes in connectivity patterns.
> **Temporal shifts** are driven by continuous evolution, including changes in the descriptions of classes and alterations in node features over time.
>
> We introduce temporal Weisfeiler-Lehman (WL) subtree patterns to effectively model these two types of distribution shifts into our proposed method. As outlined in Definition 1, we have separately proposed the structure and temporal components within the temporal WL subtree patterns.
> These patterns form the foundation for the LTDO module, which provides a computationally efficient approach for quantifying and adapting to multi-domain distribution shifts, ensuring robustness and adaptability in open temporal graph learning.
>
>
> [1] Kaituo Feng, Changsheng Li, Xiaolu Zhang, and Jun Zhou. Towards open temporal graph neural
> networks. In The Eleventh International Conference on Learning Representations, ICLR, 2023.
>
> [2] Yuan, Q., Guan, S. U., Ni, P., Luo, T., Man, K. L., Wong, P., & Chang, V. (2023). Continual graph learning: A survey. arXiv preprint arXiv:2301.12230.
>
> [3] Salehi, M., Mirzaei, H., Hendrycks, D., Li, Y., Rohban, M. H., & Sabokrou, M. (2021). A unified survey on anomaly, novelty, open-set, and out-of-distribution detection: Solutions and future challenges. arXiv preprint arXiv:2110.14051.

---

### Official Review · Reviewer_h3bq · 2024-11-04

**Soundness:** 3
**Presentation:** 2
**Contribution:** 3
**Rating:** 5
**Confidence:** 3

**Summary:**

N-ForGOT proposes a novel solution to Open Temporal Graph Learning (OTGL). Specifically, N-ForGOT consists of 2 plug-in modules: Temporal Interclass Connectivity Regularization Module (TICR), which mitigates catastrophic forgetting when new classes emerge in future timestamps, and Localized Temporal Graph Discrepancy Optimization (LTDO) that learns task-invariant representations, enhancing the generalizability for models across different tasks.

**Strengths:**

1. N-ForGOT achieves competitive empirical results and conducts a comprehensive empirical evaluation setting that consists of diverse real-world datasets and robust evaluation metrics that assess the generalizability and model’s robustness against forgetting.

2. Overall comprehensive presentation and reproducibility details, as the paper presents a clear diagram describing the method, pseudo-code for training realization, and detailed report of experimental settings, including hyperparameters and computational resources.

**Weaknesses:**

1. The mathematical notations are hard to follow in Section 4.1, as some notations are not associated with a formal definition or description.
In line 238:
>  … types for the class $i$ are denoted by $\mathbf{p}^{(i)} \sum_{j \in \mathcal{S}_i} g(x_j)$.
Before line 238, or after the definition of prototypes for a class, the definition of $\mathcal{S}_i$ is not found. Does $\mathcal{S}_i$ have the same definition as $\mathbf{S}_i$ defined in line 259? If so, I think authors should declare this and move the definition to the right behind (or before) the definition of prototypes for the class $i$ in line 238.

In line 241, what is the definition of $\mathbf{p}^{i}_j$?

In line 242, for the definition of $I(i, j)$:
> … is an indicator function used to represent whether a category $n$ appears in a task $j$
Why does $I(i, j)$’s definition depend on $n, j$?

In lines 252-253:
> … $\mathbf{p}\_{k - 1}$ is obtained from the class prototype matrix …
What is the definition of $\mathbf{p}{k - 1}$? Does it have the same meaning with the class prototype matrix $\mathbf{P}{k}$ (defined in lines 239-240 but with a different index ($k - 1$ instead of $k$)? If so, authors should revise $\mathbf{p}\_{k - 1}$ to $\mathbf{P}\_{k - 1}$.

2. What is the role of the bounding risk (presented in Theorem 1)? Does the bounding inspire any component of N-ForGOT? Or do any implications from the bound motivate the proposed method? If so, the authors should state this in the paper.

3. As authors claim that TLDO helps cultivate task-invariant representations that are robust to structural and temporal shifts, I suggest authors present a plot or illustration that shows how the distributional shifts (computed by MMD) change from task-to-task.

4. Are there any constraints for $\alpha$ and $\beta$ (defined in Eq. 9)? For better reproducibility, I suggest authors report these hyperparameters used for training N-ForGOT.

**Questions:**

Please see Weakness.

---

> ### Author Response · Authors · 2024-11-22
>
> Thank you very much for your meticulous review of our paper. We have carefully revised all the notations and writing details as per your suggestions. We truly appreciate your valuable insights. In the following, we will address each of your comments individually.
>
> **Comment 2:**
> Thank you for your insightful comment. Theorem 1 is foundational to our method. This bounding robustly justifies the design of our two plug-in modules: the Temporal Interclass Connectivity Regularization Module (TICR), which minimizes forgetting, and the Localized Temporal Graph Discrepancy Optimization (LTDO), which manages discrepancies between task distributions. We will make sure to clarify these connections more explicitly in the paper.
> Detailed explanation as below:
> $$
> \\mathcal{R}_ {\\mathbb{G}_ k}(f_ k(\\theta_ k))  \\leq \\hat{\\mathcal{R}}_ {\\mathbb{G}_ {1:k-1}}(f_ k(\\theta_ k)) + div_ {\\mathcal{H}}(\\mathbb{G}_ {1:k-1}, \\mathbb{G}_ {k}) +  \\quad \\sqrt{\\frac{2d}{m} \\log\\left(\\frac{em}{d}\\right)} + \\sqrt{\\frac{1}{2m}\\log\\left(\\frac{1}{\\delta}\\right)} + \\xi
> $$
>
> To elaborate, Theorem 1 derives from the generalization error $ |\\mathcal{R}_ {\\mathbb{G}_ k} - \\mathcal{R}_ {\\mathbb{G}_ {1:k-1}}| $. Specifically, this error measures the deviation between the performance on the current task $ \\mathbb{G}_ k $ and the previous task $\\mathbb{G}_ {1:k-1}$. Based on Theorem 1, minimizing the terms on the right-hand side reduces this generalization error, thereby optimizing the population risk for the current task $\\mathcal{R}_ {\\mathbb{G}_ k}(f_ k(\\theta_ k))$ (in the left-hand side of Theorem 1's equation).
>
> Accordingly, we focus on minimizing $\\hat{\\mathcal{R}}_ {\\mathbb{G}_ {1:k-1}}(f_ k(\\theta_k))$ and the divergence $div_{\\mathcal{H}}(\\mathbb{G}_ {1:k-1}, \\mathbb{G}_ {k})$ (key terms on the right-hand side of Theorem 1's equation), leading to the corresponding proposal of two specific plug-in modules: TICR and LTDO.
>
> **Comment 3:**
> Thank you for your suggestion. We calculate and present the task-to-task distributional shifts, computed by Maximum Mean Discrepancy (MMD), in the following Table. These results represent the average outcomes derived from the MMD calculations on the learned node representations. For further details, we will provide log files as supplementary material.
>
> Furthermore, we have depicted the distributional shifts across different tasks using a heatmap to visually represent the structural and temporal shifts [1], which is illustrated in Figure 1(b). These shifts were quantified using the Amazon dataset. For comprehensive transparency, detailed calculations will be included in the accompanying code implementation.
>
> The data presented in the following table and the heatmap (Figure 1(b)) clearly illustrate the inherent challenges posed by structural and temporal shifts in open temporal graph learning. These visualizations show how these shifts manifest across tasks, with MMD values highlighting the extent of distributional changes as new tasks are introduced. In reference to the ablation study shown in Table 2, we found that the LTDO module significantly enhances the model's effectiveness and robustness across four datasets, particularly in the context of the distribution shifts noted.
>
> **Table: Task-to-task distributional shifts (measured by MMD)**
> | **Tasks**          | **0-1** | **1-2** | **2-3** | **3-4** | **4-5** | **5-6** |
> |-------|---------|---------|---------|---------|---------|---------|
> | **Taobao Dataset** | 4.245   | 2.228   |         |         |         |         |
> | **Yelp Dataset**   | 1.015   | 0.681   | 0.802   | 1.068   | 1.348   |         |
> | **Reddit Dataset** | 1.739   | 1.456   | 2.113   | 1.921   | 1.763   |         |
> | **Amazon Dataset** | 2.148   | 2.005   | 1.833   | 1.785   | 1.698   | 2.790   |
>
> [1] Gui, S., Li, X., Wang, L., & Ji, S. (2022). Good: A graph out-of-distribution benchmark. Advances in Neural Information Processing Systems, 35, 2059-2073.

---

> ### Author Response · Authors · 2024-11-22
>
> **Comment 4:** Thank you for highlighting this point. There is no constraint for $\\alpha$ and $\\beta$. In our experiments,  both hyperparameters $\\alpha$ and $\\beta$  are settled to 1. To provide clear insights into the impact of these hyperparameters, we conducted an additional ablation study, with the results detailed in the Table below ( A better view is available in Appendix E.5). The findings show that the model achieves greater robustness when similar weights are assigned to each plug-in module in the final loss function calculation.  Both plug-in modules are significant for managing the balance between retaining previous data distributions and adapting to new incoming data, thereby improving overall model robustness. We have included this in the experiment section.
>
> **Table: Hyperparameters Ablation Study**
> | **Hyperparameters** | **AP**   | **AF** | **BWT**  |
> |------------------------------|----------|----------|----------|
> | **$\\alpha = 1, \\beta = 1$** | **69.356**   | **-2.829**   | **-2.050**   |
> | **$\\alpha = 1, \\beta = 0.8$**| 67.807 | -6.441 | -5.979   |
> | **$\\alpha = 1, \\beta = 0.4$**| 60.848 | -12.971 | -8.056
> | **$\\alpha = 1, \\beta = 0$**  | 65.342 | -1.932 | -1.932 |
> | **$\\beta = 1, \\alpha = 0.8$**|   66.890 | -2.721 | -2.158 |
> | **$\\beta = 1, \\alpha = 0.4$**|    65.382 | -9.236 | -8.100 |
> | **$\\beta = 1, \\alpha = 0$**  |      70.268 | -6.765 | -3.975 |
>
>
> Thank you for your insightful feedback. We truly appreciate the care and attention you have given to our paper, and the detailed suggestions you have provided.  We hope that our response has addressed your concerns. We will include the above discussion in the final version of our manuscript. We sincerely appreciate your reconsideration and kindly ask you to re-evaluate our work in light of these clarifications and additions.

---

> ### Author Response · Authors · 2024-11-30
>
> We hope that these additional experiments and clarifications have sufficiently addressed your concerns. Please let us know if you have any further questions or require additional information.
>
> Thank you once again for your valuable time and input.

---

### Official Review · Reviewer_TkEf · 2024-11-07

**Soundness:** 3
**Presentation:** 3
**Contribution:** 3
**Rating:** 8
**Confidence:** 2

**Summary:**

This paper addresses the challenge of open temporal graph learning (OTGL), where temporal graph neural networks must adapt to continuously emerging new classes while preserving knowledge of existing classes. The method aims to balance preventing catastrophic forgetting while enabling generalization to new classes. The authors provide a theoretical analysis of the generalization error bounds.

**Strengths:**

Originality: The paper presents a novel perspective by jointly addressing forgetting and generalizability in temporal graphs, rather than treating them as separate problems. The LTDO module's use of temporal WL subtree patterns is particularly innovative.

Quality: The theoretical analysis is thorough, grounding the approach in PAC-Bayesian theory and providing a clear justification for the chosen architecture. The reduction in computational complexity from O(n²) to O(n) is significant.

Clarity: The paper is well-structured and clearly written. The problem motivation using real-world examples (like evolving cell phone characteristics) helps readers understand the practical implications.

Significance: The work addresses an important real-world challenge in temporal graph learning and provides a practical solution that could benefit various applications like social networks and e-commerce systems.

**Weaknesses:**

1. While the TICR module's preservation of decision boundaries is well-explained, there could be more discussion about potential trade-offs between maintaining old boundaries and adapting to new class characteristics.

2. The paper could better explain how the temporal WL subtree patterns are specifically designed to capture temporal information, as this seems crucial to the method's success.

3. The relationship between batch size and performance of the LTDO module deserves more analysis, as this could affect practical deployments.

**Questions:**

The computational complexity reduction is impressive - are there any trade-offs in terms of model accuracy when using the more efficient LTDO approach compared to traditional MMD-based methods?

---

> ### Author Response · Authors · 2024-11-22
>
> One immediate strategy to address the trade-off between efficiency and accuracy is the adaptive application of the LTDO module and traditional MMD-based methods according to the task's graph size.  As demonstrated in Figure 5, the computational overhead of traditional MMD-based methods increases significantly with the number of nodes, showing that a key factor affecting the time cost is the size of the dataset.
>
> For tasks with larger graph sizes, N-ForGOT can adaptively deploy the LTDO module to ensure efficiency without substantially compromising accuracy. Specifically, we can design a hyperparameter that dynamically adjusts the proportion of traditional and LTDO methods based on the dataset size. For instance, through a threshold-based mechanism, the model could prioritize traditional MMD-based methods for smaller graph sizes below a certain threshold to maximize accuracy. Conversely, for larger graph sizes above this threshold, the proportion of the LTDO module could be increased to enhance computational efficiency. This adaptive approach ensures an optimal balance between efficiency and accuracy, tailored to the specific demands of each task.
>
>
> Another strategy involves optimizing the structure of temporal Weisfeiler-Lehman (WL) subtree patterns by adjusting their depth and the number of nodes at the leaf level. Utilizing relatively smaller temporal WL subtree patterns could enhance the efficiency of the LTDO module, whereas more comprehensive patterns might improve its accuracy. We can consider applying methods like pruning to refine the structure of these temporal WL subtree patterns further. For example, selecting more significant nodes as leaf nodes could allow the LTDO module to achieve a better balance between accuracy and efficiency.
>
>
> Thank you for your insightful comments. We will include the discussion in our final manuscript. We will also carefully consider the weaknesses you have highlighted, as they offer valuable insights for our future work.

---

> > ### Comment · Reviewer_TkEf · 2024-11-26
> >
> > I acknowledge the efforts made by the authors.

---

### Meta-Review · Area_Chair_LBwg · 2024-12-18

**Metareview:**

### Summary
The paper proposes N-ForGOT, a framework for open temporal graph learning (OTGL), which enables temporal graph neural networks to adapt to new classes while mitigating catastrophic forgetting. N-ForGOT introduces two modules: Temporal Inter-Class Connectivity Regularization (TICR) to preserve decision boundaries and prevent forgetting, and Localized Temporal Graph Discrepancy Optimization (LTDO) to enhance generalizability by learning task-invariant representations. The authors provide a theoretical analysis of generalization error bounds and conduct experiments on public datasets to demonstrate the method's effectiveness.

### Strengths
- The paper addresses OTGL with a novel perspective, simultaneously tackling catastrophic forgetting and generalization issues through two well-designed modules (TICR and LTDO).
- A solid theoretical analysis, including generalization error bounds, grounds the proposed method.
- The method demonstrates strong performance across multiple benchmarks, with clear details provided on experimental settings and training processes.

### Weaknesses
- Open-Set Learning Assumptions: The paper’s definition of "open-set" appears limited to new classes within a predefined set of tasks, raising questions about its applicability to unseen classes beyond these tasks.
- Limited Distribution Shift Analysis: While distribution shifts are mentioned, the focus seems restricted to class label changes. Other forms of shifts, such as feature or interaction dynamics, are not thoroughly modeled or discussed.


The paper addresses a critical and underexplored challenge in open temporal graph learning (OTGL) by introducing a well-structured framework, N-ForGOT, that effectively balances generalizability to new classes and the prevention of catastrophic forgetting. The proposed TICR and LTDO modules are thoughtfully designed and grounded in solid theoretical analysis, including generalization error bounds. Empirical experiments on public datasets demonstrate strong performance, highlighting the practical value and robustness of the approach. While there are minor issues with notation clarity and open-set assumptions, these do not significantly detract from the paper’s overall contributions and potential impact.

**Additional Comments On Reviewer Discussion:**

Major concerns raised by the reviewers are summarized in the above weaknesses. The authors successfully addressed most of the concerns.

---

### Decision · Program_Chairs · 2025-01-22

Accept (Poster)